# Weakly Supervised Learning for Whole Slide Lung Cancer Image Classification

**Xi Wang**[1], **Hao Chen**[1,2]\*, **Caixia Gan**[3], **Huangjing Lin**[1], **Qi Dou**[1],
**Qitao Huang**[3], **Muyan Cai**[3]\* , **and Pheng-Ann Heng**[1]

[1]Department of Computer Science and Engineering, The Chinese University of Hong Kong
[2]Imsight Medical Technology Inc, China
[3]Sun Yat-sen University Cancer Center, State Key Laboratory of Oncology in South China
Collaborative Innovation Center for Cancer Medicine, Guangzhou, China

## Abstract

Histopathology image analysis serves as the gold standard for diagnosis of cancer and is directly related to the subsequent therapeutic treatment. However, pixel-wise delineated annotations on whole slide images (WSIs) are time-consuming and tedious, which poses difficulties in building a large-scale training dataset. How to effectively utilize available whole slide image-level label, which can be easily acquired, for deep learning is quite appealing. The main barrier on this task is due to the heterogeneous patterns in fine magnification level but only the WSI-level labels are provided. Furthermore, a gigapixel scale WSI can not be easily analysed due to the immeasurable computational cost. In this paper, we propose a weakly supervised approach for fast and effective classification on whole slide lung cancer images. Our method takes advantage of a patch-based fully convolutional network for discriminative block retrieval. Furthermore, context-aware feature selection and aggregation strategies are proposed to generate globally holistic WSI descriptor. Extensive experiments demonstrate that our method outperforms state-of-the-art methods by a large margin with accuracy of $97.1\%$. In addition, we highlight that a small number of available coarse annotations can contribute to further accuracy improvement. We believe that deep learning has great potential to assist pathologists for histology image diagnosis in the near future.

## 1 Introduction

Lung cancer is the leading cause of cancer death in both men and women in the US. Appropriate treatment for lung cancer patients primarily depends on the type of lung carcinoma, such as small cell lung cancer (13%) or non-small cell lung cancer (84%). The most common non-small cell lung cancer can be divided into several main types that named based upon the tumour cells, like adenocarcinomas and squamous cell carcinomas. A range of diagnostic tests can be used to diagnose lung cancer, including chest X-ray, computerized tomography (CT), and needle biopsy. Among these approaches, histopathological image analysis serves as the gold standard for lung cancer diagnosis. Classification of carcinoma types and assessment of aggressiveness are indeed essential for following targeted treatment. In clinical practice, carcinoma is routinely identified by experienced pathologists through checking of tissue slide stained with hematoxylin and eosin (H&E) under high-power microscopy, which is a labour-intensive and time-consuming task, as it demands pathologists to look through large swathes of normal tissue regions to eventually recognize the malignant area. In addition, lots of mimics share similar appearance with cancer regions, which should be distinguished carefully.

---

\*Corresponding authors: `hchen@cse.cuhk.edu.hk`; `caimy@sysucc.org.cn`

1st Conference on Medical Imaging with Deep Learning (MIDL 2018), Amsterdam, The Netherlands.

Therefore, automated analysis technique is highly demanded in pathological field, which would considerably ease the workload, speed up the diagnosis and facilitate in-time treatment.

During the last decade, many histopathology image recognition tasks have been well studied by researchers, e.g., mitosis detection in breast cancer histopathology images [1, 2], gland instance segmentation from colon images [3], nuclear atypia scoring for breast cancer assessment [4]. However, these histopathology images employed for study fall into the type of region of interests (ROIs), which are deliberately selected by experienced pathologists with a much smaller size (e.g., $1000 \times 1000$) from whole slide images (WSI). Patch-level label or even pixel-wise annotation mask can be feasibly provided by pathologists for designing effective algorithms at the training phase. Therefore, most of the approaches can be categorized into fully supervised learning methods inherently.

With the advent of whole slide scanning techniques, dramatically increasing interest has been shown on whole slide image analysis which is much more challenging than ROI-level analysis. For example, a gigapixel whole slide image contains more than billions of pixels (e.g., $74,000 \times 76,000$) on the highest resolution level, which poses great challenges to image-level classification. Downsampling WSIs into thumbnails is not feasible as a lot of intrinsic information and significant details would be lost. Alternatively, it is more reasonable to perform analysis on small patches with fine details cropped from high-resolution WSIs, which is similar to ROI-level analysis, but in a much larger scale as thousands of patches should be taken into consideration for WSI classification.

Regarding the high-level whole slide image classification task, due to lacking detailed annotations of cancer regions, previous studies applied domain-specific hand-crafted features to depict morphological, texture and statistic property of malignant tumour [5, 6, 7] along with unsupervised methods (e.g., K-means) and feature embedding ahead of classification by a standard classifier, such as Adaboost or a support vector machine (SVM) [8]. In addition, various weakly supervised methods, like multiple instance learning (MIL) [9, 10, 11, 12], have been adopted to address this problem by automatically extracting refined valuable information from coarse-labeled patches. Hand-crafted features (e.g., colour histogram, local binary pattern and SIFT) are also extensively studied in these MIL methods [9, 10, 11], which actually require considerable efforts to design and validate. Recently, a multi-instance learning framework [13] utilized sparse labels for supervised training to classify whole mammogram of breast cancers with size about $3000 \times 2000$. However, this idea can not easily generalize to WSIs since latter one is along with a much larger resolution. In the Camelyon Grand Challenge 2016 (CAMELYON16) [14, 15], hundreds of carefully annotated WSIs are provided to train a fully supervised deep neural network for automatically detecting metastatic breast cancer in WSIs. However, acquisition of careful annotation of whole slide tissue image in a large scale is fairly prohibitive, if not impossible, in practice as it usually takes several hours to well annotate each WSI for a specialized pathologist.

Therefore, it would be appealing to train the cancer region detector with a minimum annotation (e.g., image-level labels) [16, 17], which can be more easily acquired in the clinical practice. Recently, an EM-based method proposed by [16] is the first to combine patch-based convolutional neural network (CNN) with supervised decision fusion. Initially, an EM-based method with CNN is used to identify discriminative patches in WSIs, then a count-based feature fusion model performs the image-level prediction. Although this method has been effective by evaluation on two WSI datasets, it merely marginally exceeds fundamental methods reported in [16] at the cost of huge computation on iteration of training and inference.

In order to overcome these challenges, in this paper we propose a weakly supervised learning method for fast and effective classification of whole slide lung cancer images with a minimum annotation from pathologists. We first take advantage of fully convolutional network for efficient prediction and its representative ability to extract features. Then spatial information within WSIs is taken into consideration in feature selection. Extensive experiments demonstrate our context-aware block selection strategy and WSI feature aggregation from multiple instances can guarantee the quality of holistic feature representation of WSIs, which boost the performance dramatically compared to the state-of-the-art methods.

## 2 Method

Figure 1 shows the architecture of our proposed method which consists of three parts. The first part is a patch-based CNN that aims to predict the cancer likelihood from WSIs, referred as *Discriminative*

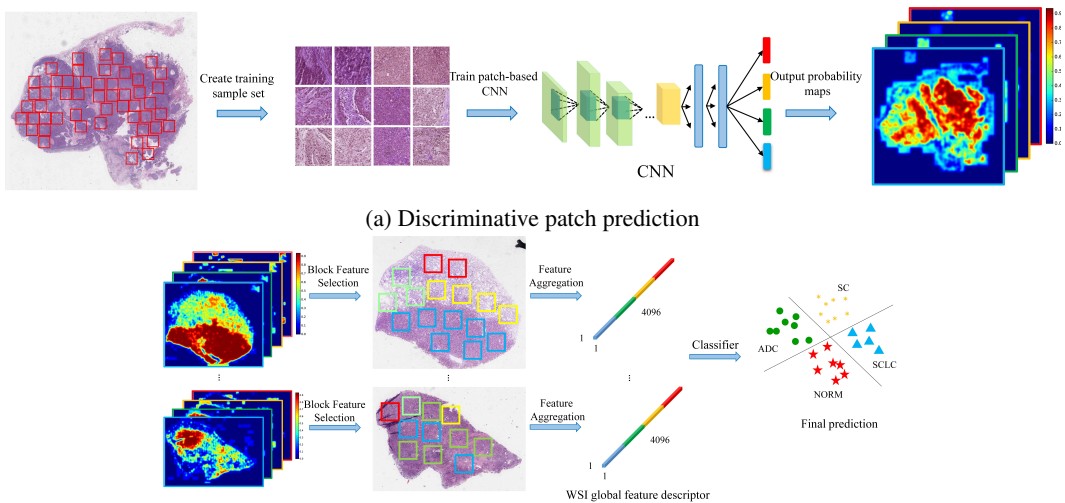

(a) Discriminative patch prediction

(b) Context-aware feature selection and aggregation

Figure 1: An overview of the proposed method. (a) The discriminative patch prediction. A patch-based CNN is used to find discriminative regions. (b) The context-aware feature selection and aggregation. By imposing spatial constraint, features from discriminative blocks are selected and aggregated for Random Forest classification.

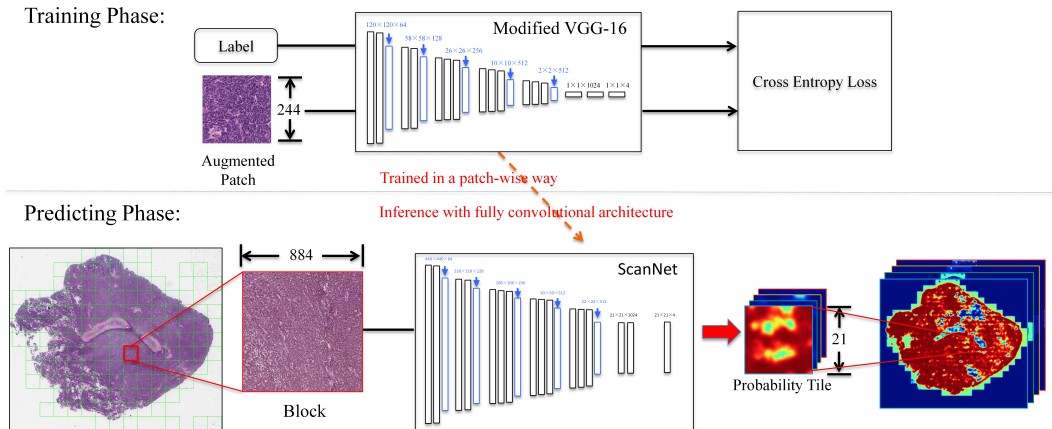

Figure 2: The illustration of fast patch prediction with ScanNet.

*Patch Prediction*. In the second part of *Context-aware Block Selection*, the spatially contextual information is taken into consideration when selecting features from these retrieved blocks. Finally, we aggregate features from multiple representative instances, hence each WSI can be represented by a global feature descriptor that summarizes the most indicative information. Afterwards, the global feature descriptor is fed into a standard Random Forest (RF) classifier for WSI-level prediction. This process shares the similar idea of "Vocabulary-based Paradigm" [8] in embedded-space MIL, referred as *WSI feature aggregation and classification*.

## 2.1 Discriminative Patch Prediction

### Fast Fully Convolutional Network

We adopt a modified fully convolutional network (FCN), ScanNet [14], as the patch-prediction model. This framework can be flexibly trained with exhaustive data augmentation in a patch-wise way while it can leverage the efficiency of FCN architecture in prediction phase, as illustrated in Figure 2. The architecture of ScanNet is based on a modified VGG-16 [18] network by replacing the last three fully

connected layers with fully convolutional layers, which can enjoy the transferred features learned from a large set of nature images [19]. Additionally, all the padding operations of convolutional layers in the standard VGG-16 are removed to avoid the boundary effect of FCN predictions. Based on this modification, the ScanNet can fast predict region blocks in arbitrary size by leveraging the efficiency of FCN architecture as long as the GPU memory allowed. Finally, all probability tiles generated from blocks are stitched together to form the probability map of WSIs.

**Weighted Loss Function for Weakly Supervised Learning**

There are at least two challenges for fully supervised learning of WSI analysis. First, it is quite difficult as well as tedious to obtain accurately pixel-wise annotations. Second, there exist ambiguous regions that can not be well distinguished, even for histology experts. However, making use of a large number of available image-level labels and a small number of coarsely annotated WSIs can be feasible in the clinical practice. In this study, we are the first to explore weakly supervised learning on WSI classification with image-level labels as well as a small number of coarsely annotated abnormal regions. We ask the pathologists to annotate the abnormal regions in a scratch way by drawing a polygon as shown in Figure 5 of Appendix. As the annotation is quite coarse, not all annotated areas are precisely occupied by tumour, and vice versa. Hence it is not safe to take all annotated regions as positive patches, and non-annotated counterparts as negative ones at the training stage. A more reasonable way is to emphasize more weights on these annotations as they carry more confidence for manifestation of being carcinoma. Specifically, we train a patch-based CNN by minimizing the following weighted cross-entropy loss function:

$$\mathcal{L} = \sum_{x \in \mathcal{M}} \sum_{c=1}^{C} -\alpha y_c \log P(q_c = 1 | x; \mathcal{W}, b) + \sum_{x \notin \mathcal{M}} \sum_{c=1}^{C} -y_c \log P(q_c = 1 | x; \mathcal{W}, b) + \lambda \|\mathcal{W}\|_2^2 \quad (1)$$

where $\theta = \{\mathcal{W}, b\}$ denotes the parameters of our CNN model, $P(q_c = 1 | x)$ is the output probability for the $c$-th class given the input sub-window $x$, and $y_c$ corresponds to the WSI-level label. $C$ is the total number of classes. $\mathcal{M}$ denotes the coarse annotation mask set. $\alpha$ is the balance weight between annotated region classifier and non-annotated classifier, which was set as 2 in our experiment. $\lambda$ controls the tradeoff between the data loss term and regularization term.

## 2.2   Context-aware Block Selection

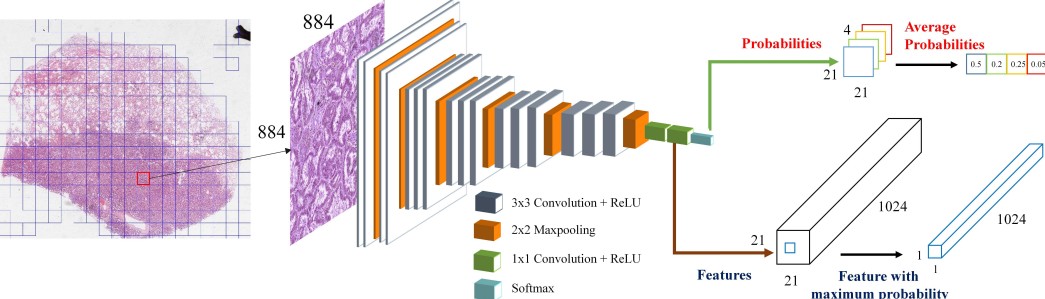

Figure 3: Context-aware feature selection. A block, larger than a patch, with size $n \times n$ is regarded as discriminative only if its average probability exceeds a certain threshold $\tau$. Then features are extracted from each block according to different strategies, e.g., *MaxFeat*.

We hypothesize that patch with higher probability for a specific class is more likely true. Thus, the feature extracted from such region would be more reliable than from that with lower probability. Previous study [16] utilized all discriminative patches and the corresponding features. However, it will lead to feature redundancy during inference as CNN densely slides over the WSI and many patches share the overlapped region with their neighbours. On the other hand, the heterogeneity of histopathological characteristics exists in WSIs. Hence, there would be outliers or mimics that have high probability, resulting in negative effect on the quality of subsequent WSI holistic feature representation, which eventually degrade the performance of image-level classification.

In order to tackle above issues, rich contextual information is taken into consideration for better feature selection as illustrated in Figure 3. Here a *block* refers to be a larger grid that consists of a

number of overlapped patches. Then a whole slide image can be regarded as a composition of many blocks. In general, tumour area has a larger size than a patch does, resulting in high probability scores appearing in a concentrated region. In other words, the average probability of such region would certainly be high. On the contrary, if an outlier carrying a high probability value falls in a normal tissue block, it is easy to be filtered out due to the low average probability of this block. Briefly, We denote a block as $B = \{I_{1,1}, I_{1,2}, ..., I_{n,n}\}$, where $I_{i,j}$ is the patch located at the $i$-th row and $j$-th column in block $B$. Each patch $I_{i,j}$ generates a probability vector $p_{i,j} = \{p_{i,j,1}, p_{i,j,2}...,p_{i,j,C}\}$ where $p_{i,j,c}$ means the probability score for the $c$-th class. For each class $c$, average probability within a block is calculated by $\bar{p}_c = \frac{1}{n^2} \sum_{i,j} p_{i,j,c}$, which is used to identify the discriminative blocks if exceeding a certain threshold $\tau$, which was set as 0.3 in our experiment.

### 2.3 WSI Feature Aggregation and Classification

A good holistic feature descriptor is essentially required for classifying each WSI. Intuitively, it should integrate global information from all cancer types and non-cancer type. We call them positive and negative evidence. Specifically, the positive evidence can well support the existence of cancer class that is consistent with the ground truth. In contrast, the negative evidence can manifest the absence of any other classes. The general procedure to obtain the holistic representation of WSI consists of three-stage feature aggregations. First of all, perform feature aggregation within each discriminative block, which can be regarded as block-level feature fusion $F_b$. The outcome of this phase is termed *block descriptor* which is supposed to represent one block region. Afterwards, fuse information among blocks to obtain the specific class feature, which is named *class descriptor*. It can support the existence or absence of the class. Eventually, all class descriptors are concatenated together to interpret the WSI, which is referred as *global descriptor*.

At the first stage, there are several different strategies to aggregate features within a block. The first approach is called *MaxFeat*, which takes the feature $f$ of the patch with the highest probability as the block descriptor:

$$D_b = f_{max_{(i,j)} p_{(i,j)}} \tag{2}$$

The second strategy is the fusion of all patch-level features with equal contribution, called *AvgFeat*:

$$D_b = \frac{\sum_{i,j} f_{i,j}}{n^2} \tag{3}$$

Similarly, another strategy is to consider all features within the same block but the contribution of each individual patch-level feature to the block descriptor is directly proportional to its probability score, referred as *WeightFeat* as following:

$$D_b = \sum_{i,j} p_{i,j} f_{i,j} \tag{4}$$

In the second stage, we simply average all the discriminative block descriptors to get the class descriptor $D_c$:

$$D_c = \frac{\sum_b D_b}{N_{block}} \tag{5}$$

where $N_{block}$ denotes the number of discriminative blocks for the class $c$. In such a way, each class descriptor has the same dimension. Then, all the class descriptors are concatenated together in order to generate the global descriptor $D_g = \{D_1, ..., D_C\}$. The detailed process of feature selection and aggregation is illustrated in Figure 3 and Figure 1(b) accordingly. The block size $n = 884$ is determined by cross-validation in our study. Each block would output probability maps with size $21 \times 21 \times C$. Note that each class is distinguished by different colour in Figure 1 and Figure 3 for clear illustration. Finally, the global descriptors with good holistic representations are fed into a standard Random Forest classifier for WSI-level prediction.

## 3 Experimental Results

### 3.1 Dataset and Preprocessing

We conducted extensive experiments on 871 digitalized histology WSIs with lung carcinomas and 68 WSIs of healthy subjects from Sun Yat-Sen University Cancer Center. The 871 WSIs are

diagnosed into three fine-grained categories of lung cancer, i.e., Squamous cell Carcinoma (SC), Adenocarcinoma (ADC) or Small Cell Lung Carcinoma (SCLC). Within this set, 59 images are annotated by a panel of experienced pathologists. Due to great efforts required for pixel-wise annotation of WSI, only coarse annotations of carcinoma regions are collected, as illustrated in Figure 5 of Appendix. We consider these 59 annotated images as D1. Besides, the rest 812 cancer images only carry the WSI-level labels, which are further split into 642 (D2) and 170 (D4) images for training and testing, respectively. Analogously, non-cancer (Normal) WSIs are also divided into two parts containing 53 (D3) and 15 (D4) images for training and testing accordingly, as shown in Table 1.

All the images are obtained using a Leica Aperio AT2 scanner at a 40X magnification with $0.25\mu m$/pixel resolution, and are stored in multiple zoom levels (3 or 4) with a pyramid-like structure. At the finest magnification, the whole slide images come with an average size of $74,000 \times 76,000$ pixels. Considering that processing the images at this finest magnification level would be intractable due to the huge computations, we down-sampled each image by a factor of 4 to the resolution of $1\mu m$/pixel at pre-processing. An effective segmentation algorithm of OTSU's method [20] is employed to remove a large proportion of non-informative white background. During training, we adopt the same data augmentation techniques in [14] to enrich training dataset, including rotation, translation, flipping and color jittering.

Table 1: Data distribution in our dataset.

|  |  | Carcinoma | SC | ADC | SCLC | Normal |
|---|---|---|---|---|---|---|
| Training | D1 | 59 | 21 | 20 | 18 | - |
|  | D2 | 642 | 267 | 293 | 82 | - |
|  | D3 | - | - | - | - | 53 |
| Testing | D4 | 170 | 73 | 77 | 20 | 15 |

## 3.2 Experimental Settings

**Configuration of Training Datasets**

1. *M1*: In this experiment, D1 (59 cancer WSIs) and D3 (53 non-cancer WSIs) are used for patch-based CNN training. All patches extracted from D1 and D3 only convey the WSI-level labels. During inference, CNN scans the training WSIs and outputs the patch-wise probabilities as well as the features from the last convolutional layer.

2. *M2*: It is quite similar to M1 except that the weighted loss function is employed during training, which gives higher penalty to patches extracted from annotated regions.

3. *M3*: The training data consists of D1, D2 and D3 (i.e., 701 cancer images and 53 non-cancer images). Note that coarse annotation masks are not utilized during CNN training.

4. *M4*: Analogous to M3, M4 utilizes the same training dataset: D1, D2 and D3. The only difference is that weighted loss function is utilized on coarse annotation regions.

**Configuration of Feature Aggregation Methods**

We compare different feature aggregation methods to obtain the image-level prediction of the whole slide image. *MajorityVoting*: We inference the CNN on the testing WSI and obtain a score map. The prediction of each location votes to the four classes and we take the majority vote category as the prediction for the image. *AveragePooling*: We calculate the average probability of the locations on the test WSI score map for each class channel. The category with the highest average probability is taken as the image-level prediction. *MaxPooling*: We select the maximum probability of the locations on the test WSI score map for each class. The category with the highest max pooling probability is taken as the image-level prediction. *Count-based RF*: We count the numbers of all cancer types and non-cancer type prediction in test WSI score map to form a prediction histogram of classes. The four-bit histogram is fed into an RF classifier for image-level prediction. *Component-based RF*: For each test score map, the connected component with the largest area for each class is chosen as the region of interest. Then, we obtain features of these ROIs including maximum probability, average probability, area, eccentricity, convex area, orientation, extent, equivalent diameter, solidity, major axis length, minor axis length and perimeter. Finally, an RF classifier takes the feature vector as input to get the final prediction. *CNN-AvgFeat-based RF*, *CNN-WeightedFeat-based RF*, *CNN-MaxFeat-*

*based RF*: Our proposed feature aggregation methods in Section 2.3. *AvgFeat*, *WeightedFeat* and *MaxFeat* are used respectively to obtain block descriptor. After feature aggregation, the RF gives the WSI-level prediction based on the global descriptor.

### 3.3 Quantitative Evaluation and Comparison

We employ the accuracy as the evaluation criteria for our multi-class WSI classification task. The extensive experimental results are listed in Table 2. We combine different training strategies with different feature selection and aggregation methods as ablations studies to evaluate the contribution of each crucial component within our framework.

As for the first three simple prediction strategies, *MajorityVoting*, *AveragePooling* and *MaxPooling*, the performance is not very competitive. The reason behind would be that these methods only rely on instances whereas in lack of effective holistic information of WSI. Besides, there is no gain of improvement by adding more training samples as these methods are quite sensitive to outliers, which eventually degrades the accuracy. With respect to *Count-based RF* and *Component-based RF*, there is a considerable boost on accuracy. They integrate information from instances and create a global feature vector for the second-stage classifier, which are more robust to output the WSI-level prediction. Inspiringly, context-aware CNN feature selection and aggregation methods, i.e., *CNN-AvgFeat-based RF*, *CNN-WeightedFeat-based RF* and *CNN-MaxFeat-based RF*, outperform the preceding methods by a large margin, where not only the feature of multiple instances is used, but also rich spatial information is also taken advantage of. It is validated that the features learned by convolutional neural network are more representative than count-based histogram and component-based features. We can observe that *CNN-MaxFeat-based RF* classifier achieves the best result among all WSI-level prediction strategies based on the setting of M4. In addition, we can also notice that a small number of coarsely annotated masks (M4) can contribute to the accuracy improvement compared with that without (M3).

Table 2: Results from different training datasets and feature selection methods

| Method | M1 | M2 | M3 | M4 |
|---|---|---|---|---|
| MajorityVoting | 0.708 | 0.719 | 0.665 | 0.697 |
| AveragePooling | 0.730 | 0.735 | 0.676 | 0.703 |
| MaxPooling | 0.530 | 0.681 | 0.616 | 0.627 |
| Count-based RF | 0.770 | 0.783 | 0.875 | 0.930 |
| Component-based RF | 0.748 | 0.759 | 0.909 | 0.935 |
| CNN-AvgFeat-based RF | 0.786 | 0.812 | 0.928 | 0.955 |
| CNN-WeightedFeat-based RF | 0.767 | 0.858 | 0.932 | 0.960 |
| CNN-MaxFeat-based RF | 0.732 | 0.824 | 0.953 | **0.971** |

We further implement two weakly supervised learning methods which are the state-of-the-art in WSI analysis for comparison. One is the EM-based method with CNN followed by a supervised decision fusion model proposed by [16], and the other is a CNN activation feature-based method [21], which takes advantage of the CNN pre-trained on ImageNet to extract features from all patches in each WSI. In [21], 3-norm pooling is used to aggregate all features and feature dimension reduction is applied to remove irrelevant features. From Table 3, we can see our methods overwhelm these approaches significantly. This success primarily owes to the representative features from CNN in conjunction with context-aware feature selection and aggregation strategy since the quality of the global descriptor representing a WSI is crucial for WSI-level classification.

Table 3: Comparison with state-of-the-art methods on WSI image classification

| Method | Accuracy |
|---|---|
| Pretrained-Feature-Norm3 [21] | 0.874 |
| EM-CNN-SVM[16] | 0.914 |
| M3-CNN-MaxFeat-based RF (Ours) | 0.953 |
| M4-CNN-MaxFeat-based RF (Ours) | **0.971** |

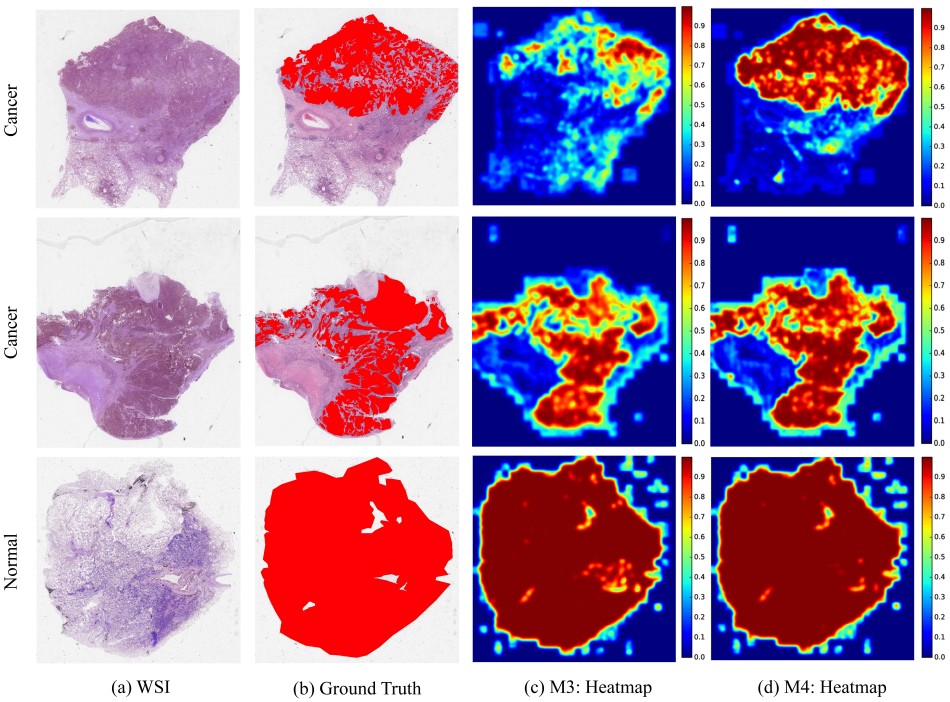

(a) WSI      (b) Ground Truth      (c) M3: Heatmap      (d) M4: Heatmap

Figure 4: Visualization of discriminative region detection.

### 3.4 Qualitative Evaluation

Albeit our ultimate task is not tumour detection, our method can achieve such a goal simultaneously by retrieving the most discriminative regions. We invite a specialized pathologist to delineate the discriminative regions elaborately on a few testing WSIs. The results are depicted in Figure 4. The Figure 4(a) and 4(b) show the WSI and ground truth accordingly, followed by heatmaps generated on M3 and M4 in Figure 4(c) and 4(d), respectively. Note that the first two rows are cancer cases, and the last row is a normal case. The red regions in 4(b) denote carcinoma regions in cancer cases or normal tissue in normal case. Clearly, despite that the patch-based CNN could learn discriminative patterns from WSIs without the aid of annotation information, it sometimes could not find the most discriminative regions, which might lead to feature deficiency for the positive evidence. On the contrary, robustness is clearly improved by adding only a handful of annotated WSIs. The heated regions found by M4 is more consistent with the annotation from the pathologists. These visualization results validate that our system makes the diagnosis decisions based on real discriminative regions.

## 4 Conclusion

In this paper, we propose a weakly supervised learning technique to address whole slide lung cancer image classification problem with minimum annotation information. The adopted fully convolutional network can make efficient prediction and provide representative features. In addition, we utilize context-aware feature selection and aggregation strategy to obtain a good holistic feature representation of WSI, which is quite effective for the WSI-level prediction. Extensive experiments show the superiority of the proposed method which outperforms state-of-the-art methods by a large margin. These results suggest that the deep learning-based approach can be broadly applied in histopathology image field to assist histopathologists tackling clinical tasks.

## 5 Acknowledgments

This project is supported by the Hong Kong Innovation and Technology Commission, under ITSP Tier 3 (Project number: ITS/041/16).

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

# Appendix

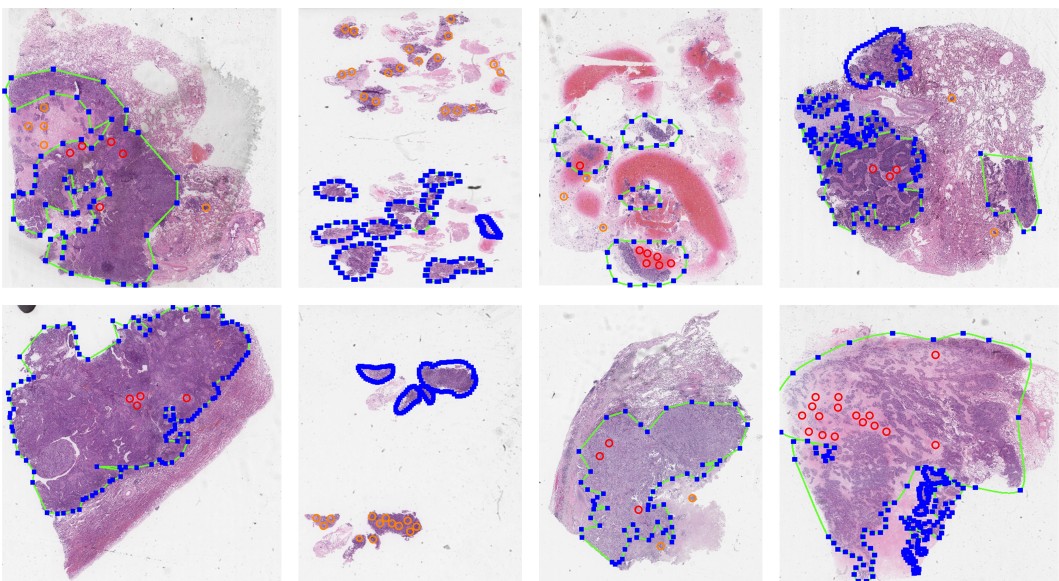

Figure 5: Illustration of coarse annotations by experienced histopathologists. Green lines with blue dots are the coarse annotations from histopathologists. Note that orange circles denote cancer regions that are not annotated while red circles indicate non-cancer regions that are enclosed in annotations.

