# OpenReview forum: "Weakly Supervised Learning for Whole Slide Lung Cancer Image Classification"
_MIDL.amsterdam/2018/Conference — MIDL 2018 Oral_

### Review · AnonReviewer1 · 2018-05-08
**nice application study, not sure about technical innovation**

**Rating:** 2
**Confidence:** 2

**Review:**

An algorithm for histopathology classification is presented that fuses both image-level labels and coarsely annotated tumor regions (in some scans). A convolutional architecture for predicting global labels is learned. Some pre-final features are used to generate spatial probability maps using an external classifier (here: random forest).

Pro: The applications are well presented, authors describe what they are doing and why. Experiments are extensive, results are promising and may well generalize beyond the specifics of this application.

Con: I have the impression the prior work can be extended significantly, as the approach follows - in many aspects -  a common modeling strategy with global labels for weakly supervised learning. As such, I would see the merits on this paper rather on the application itself than on the technical specifics of the learning algorithm.

**Special Issue:**

No

---

> ### Comment · ~Xi_Wang1 · 2018-05-14
> **reply to reviewer 3**
>
> Thanks for the comments. We agree with the reviewer that whole slide image application is quite important. In clinical, histopathology usually serves as the gold standard for cancer diagnosis. However, the examination of a single whole slide image is very tedious and time-consuming even by an experienced pathologist. Especially in some countries, like China, the supply of pathologists falls short of demand considerably. Therefore, automated analysis of whole slide images is highly demanded in clinical which could significantly ease the workload and facilitate the in-time diagnosis.
> However, our method is not only applying to the lung cancer image classification, but also advanced several methodological improvements. First of all, our method can achieve fast predictions by taking advantage of fully convolutional network, while other methods work in patch-wise fashion to predict probabilities which are extremely time-consuming. Efficiency is undoubtedly critically important for whole slide image analysis. Secondly, we design a weighted loss function to utilize both image-level labels and a limited coarsely annotations, which is superior to previous methods where only image-level labels are used. More importantly, we propose several context-aware feature selection and aggregation strategies to obtain global descriptor to represent the whole slide image for second-stage image-level classifier, which are more robust than other simple feature aggregation approaches adopted in the previous work. All in all, our method makes further progress on this whole slide image application.

---

### Review · AnonReviewer3 · 2018-05-08
**This paper proposes a weakly-supervised learning method to address whole slide lung cancer image classification problem with minimum annotation information. Besides, it adopts context-aware feature selection and aggregation strategy to obtain a good holistic feature representation of whole slide images (WSIs) to improve the WSI-level prediction performance.**

**Rating:** 4
**Confidence:** 3

**Review:**

(1) The paper is well-written and easy to follow.

(2) The proposed method is technically sound and could be potentially used to assist pathologists for histology image diagnosis.

(3) The authors claim that the proposed method can achieve fast and effective classification on whole slide lung cancer images, while there is no clue to show the “fast” property of the proposed method. The comparison of computational complexity of competing method and corresponding analysis in detail should be helpful.

(4) The global descriptors produced by the proposed method are further fed into a RF classifier. Why not consider an end-to-end classification strategy?

(5) Some details are missing. For example, the description in Sec. 3.1 is not clear: "the rest 812 cancer images only carry the WSI-level labels, which are further split into 642 (D2) and 170 (D4) images..." Is there any criterion to perform this kind of split?


**Special Issue:**

No

---

> ### Comment · ~Xi_Wang1 · 2018-05-14
> **reply to reviewer 2**
>
> (3) Thanks for the suggestion. We will revise our paper accordingly and add the computational complexity analysis in the final version. The networks used in the two prior works, EM-CNN-SVM and Pretrained-Feature-Norm3, are not fully convolutional networks. Hence, at inference phase, they work in patch-wise fashion to output probabilities or features, which is extremely time-consuming. On the contrary, our fully convolutional network can take arbitrary size patch as input, like 2,868 x 2,868 (determined by the capacity of GPU memory). By such a way, out network can process a whole slide image more than hundreds of times faster than traditional patch-wise classification framework with the same stride. In reality, it takes about 100 seconds to process one whole slide image with dimension of 20,500 x 19,500 (1.0 μm/pixel).
> (4) In the previous work, we considered to find the suitable threshold value and the size of block at cross-validation phase, therefore made the inference and image-level classification separately. In our following work, we would like to add CNN layers to make training and inference in an end-to-end fashion. Thanks.
> (5) In our experiments, we would like to utilize about 80% of non-annotated whole slide images for training and the other part (20%) for testing. Therefore, we randomly split these 812 cancer images into D2 and D4, respectively. All classes in training and testing dataset follow this ratio (4:1) empirically.

---

### Review · AnonReviewer2 · 2018-05-10
**Good paper, need some clarification to be stronger**

**Rating:** 4
**Confidence:** 3

**Review:**

In this paper, the author presented a patch-based FCN network for discriminative regions retrieval. Multiple aggregation and selection methods are evaluated. The paper is generally well-written and easy to understand. The results look good.

Here are major comments:
1.	The method is called “weakly supervised” which is usually used for segmentation/localization task with image-level annotation. In contrast, this work is for pure classification task aided with coarsely annotated discriminative regions. Changing it to another name may be more precise.
2.	Why not using CNN based classification at the last stage but random forest?
3.	As mentioned coarse annotation has errors, would iteratively updating the coarse annotation using the prediction/localization result help the overall performance?
4.	Would focal loss help the accuracy of SCLC?
5.	In equation 1, y_c or y_k?
6.	In Fig 2, 2x2x512 -> 1x1x1024, how this is done with 1x1 kernel without padding? So is the 22x22x512 -> 21x21x1024
7.	Would class balance weight in loss function help?
8.	What’s red in Figure 4? Normal?


**Special Issue:**

Yes

---

> ### Comment · ~Xi_Wang1 · 2018-05-14
> **reply to reviewer 1**
>
> 1. Thanks for the comment. In the field of image classification, “weakly supervised” can also be used to describe the method [1,2,3,4,5,6], such as multi-instance learning methods[1,2] on classification and localization tasks, which is similar to our underlying task. As defined in [6,7], weakly supervised learning is a machine learning framework where the model is trained using examples that are only partially annotated or labeled. Especially in computer vision, weakly supervised learning has been predominantly used for classification tasks such as image categorization and object detection [7]. In our paper, we utilize a large number of image-level label and limited amount of noisy annotated data to train our model. Therefore, our method inherently falls into the category of weakly supervised learning.
> [1]Xu, Y., Zhu, J.-Y., Eric, I., Chang, C., Lai, M., and Tu, Z., “Weakly supervised histopathology cancer image segmentation and classification,” Medical Image Analysis, Vol. 18, No. 3, 2014, pp. 591–604.
> [2]Xu, Y., Mo, T., Feng, Q., Zhong, P., Lai, M., Eric, I., and Chang, C., “Deep learning of feature representation with multiple instance learning for medical image analysis,” in “Acoustics, Speech and Signal Processing (ICASSP), 2014 IEEE International Conference on,” IEEE, 2014,pp. 1626–1630.
> [3]Durand, T., Mordan, T., Thome, N., and Cord, M., “Wildcat: Weakly supervised learning of deep convnets for image classification, pointwise localization and segmentation,” in “IEEE Conference on Computer Vision and Pattern Recognition (CVPR 2017),” , 2017.
> [4]Nguyen, M. H., Torresani, L., De La Torre, F., and Rother, C., “Weakly supervised discriminative localization and classification: a joint learning process,” in “Computer Vision, 2009 IEEE 12th International Conference on,” IEEE, 2009, pp. 1925–1932.
> [5]Xu, Y., Jia, Z., Ai, Y., Zhang, F., Lai, M., Eric, I., and Chang, C., “Deep convolutional activation features for large scale brain tumor histopathology image classification and segmentation,” in “Acoustics, Speech and Signal Processing (ICASSP), 2015 IEEE International Conference on,”
> IEEE, 2015, pp. 947–951.
> [6] Zhou, Z.-H., “A brief introduction to weakly supervised learning,” National Science Review.
> [7] Torresani, L., “Weakly supervised learning,” in “Computer Vision,” Springer, pp. 883–885,2014.
> 2. As for image-level classification, the input features are well semantically represented by the deep learning model. Therefore, the simple random forest could work efficiently and avoid over-fitting issue from the neural network (such as multi-layer perception). We also compared random forest with support vector machine (SVM) classifier. It turned out that random forest outperformed SVM on this task. In our future work, we would like to apply CNN-based classification to make the inference and image-level classification in the end-to-end fashion.
> 3. Good suggestion. For the M1 and M2 (small size of WSIs used for training), iteratively updating the coarse annotation might be very useful to improve the overall performance on testing images. With respect to M3 and M4, the improvement might be slight due to large number of training WSIs. It is worthwhile trying in the future work.
> 4. Focal loss is a dynamically scaled cross entropy loss that is designed to address the one-stage object detection scenario where there is an extreme imbalance between foreground and background classes during training. The scaling factor in focal loss decays to zero as confidence in the correct class increases which automatically down-weight the contribution of easy examples and rapidly focus the model on hard examples. In our task, there exists indeed class imbalance among SCLC and other classes. In particular, SCLC are the hard examples that cannot easily classified correctly. Hence, focal loss might be helpful to address such problem and improve the detection performance on finding discriminative regions on SCLC whole slide images. Thanks.
> 5. This is a typo. Thanks and we will revise it.
> 6. From 2x2x512 -> 1x1x1024, there is a convolutional layer with kernel size of 2x2 instead of 3x3. Hence, after convolution operation, the size of output would become 1x1 without padding.
> 7. It might be helpful. We will try it.
> 8. In Figure 4, the red region in (b) denote the carcinoma regions (the first two rows) or normal tissue (the last row) which is annotated deliberated by pathologists for evaluation. We will polish the explanation in the final version.

---

### Comment · ~Bram_van_Ginneken1 · 2018-05-18
**Selection for longlist for special issue Medical Image Analysis**

Dear authors,

Congratulations on your acceptance to MIDL! We have selected your paper on the longlist for the Medical Image Analysis Special Issue. Please read this page:
https://midl.amsterdam/special-issue-in-medical-image-analysis/
Please answer the three questions that are listed on that page about your interest in submitting to the special issue, potential overlap with other publications, and related publications.

You can post your answer here directly below on openreview.net, or mail me directly at bram.vanginneken@radboudumc.nl.

Best regards, Bram

---

> ### Comment · ~Xi_Wang1 · 2018-05-20
> **Reply to Bram**
>
> Dear Bram,
>
> Thanks a lot for the opportunity being selected for longlist for special issue Medical Image Analysis. We really appreciate this chance. Below are our answers of those questions on https://midl.amsterdam/special-issue-in-medical-image-analysis/.
>
> 1. Yes, we are very interested to be eligible for the special issue. We will enrich the content of our paper soon and submit the full manuscript before the deadline.
> 2. We promise that our MIDL paper will not be under review or under consideration elsewhere. We will certainly abide by the Elsevier rules.
> 3. There are some related publications of our author group below.
> [1] Lin, H., Chen, H., Dou, Q., Wang, L., Qin, J., and Heng, P.-A., “ScanNet: A fast and dense scanning framework for metastatic breast cancer detection from whole-slide images,” 2018, IEEE Winter Conference on Applications of Computer Vision (WACV).
> [2] Chen, H., Dou, Q., Wang, X., Qin, J., Heng, P.-A., et al., “Mitosis detection in breast cancer histology images via deep cascaded networks.” in “AAAI,” , 2016, pp. 1160–1166.
> [3] Chen, H., Qi, X., Yu, L., and Heng, P.-A., “Dcan: Deep contour-aware networks for accurate gland segmentation,” in “Proceedings of the IEEE conference on Computer Vision and Pattern Recognition,” , 2016, pp. 2487–2496.
> [4] Chen, H., Qi, X., Yu, L., Dou, Q., Qin, J., and Heng, P.-A., “DCAN: Deep contour-aware networks for object instance segmentation from histology images,” Medical image analysis, Vol. 36, 2017, pp. 135–146.
> [5] Chen, H., Wang, X., and Heng, P. A., “Automated mitosis detection with deep regression networks,” in “Biomedical Imaging (ISBI), 2016 IEEE 13th International Symposium on,” IEEE, 2016, pp. 1204–1207.
>
> Thanks again!
>
> Best regards,
> Xi Wang

---

### Decision · Program_Chairs · 2018-05-15
**Paper70 Acceptance Decision**

Oral